# An Ectopic Parathyroid Adenoma of the Retropharynx in a Patient with Primary Hyperparathyroidism and Papillary Thyroid Cancer—A Rare Case

**DOI:** 10.3390/diagnostics14010110

**Published:** 2024-01-04

**Authors:** Youngjae Lee, Wonyong Baek, Jiwoong Cho, Jeonghyun Oh

**Affiliations:** Department of Otorhinolaryngology-Head and Neck Surgery, Chosun University Hospital, Gwangju 61453, Republic of Korea

**Keywords:** primary hyperparathyroidism, papillary thyroid carcinoma, ectopic parathyroid

## Abstract

The frequency of concurrent thyroid cancer in patients with primary hyperparathyroidism (pHPT) varies. While the pathological association between thyroid and parathyroid disorders is frequently noted, the co-occurrence of parathyroid adenoma and papillary thyroid cancer is exceptionally rare. Furthermore, an ectopic parathyroid adenoma in the retropharyngeal space is exceedingly rare. Therefore, anatomical variations through the utilization of relevant diagnostic tools play a crucial role in guiding decisions pertaining to clinical manifestations, diagnostic methods, surgical interventions, and operative strategies for parathyroid tumors. We present a case of a 51-year-old female patient with papillary thyroid carcinoma in the right thyroid lobe and an ectopic parathyroid adenoma in the retropharyngeal space confirmed through surgical intervention. The elevated preoperative levels of serum calcium and parathyroid hormone, along with low serum phosphate, returned to normal ranges after surgery. This case sheds light on the unusual occurrence of an ectopic parathyroid adenoma in the retropharyngeal region within a thyroid cancer patient, providing valuable insights into the realm of thyroid malignancies.

A 51-year-old female patient with end-stage renal disease on dialysis was referred by the nephrology department due to suspected osteoporosis and the presence of osteoblastic lesions. Laboratory results indicated elevated serum calcium (10.4 mg/dL; normal range: 8.4~10.2 mg/dL), intact parathyroid hormone level (383 pg/mL; normal range: 12.0~65.0 pg/mL), and low serum phosphate (1.71 mg/dL; normal range: 2.3~4.7 mg/dL), as well as 25-OH vitamin D (13.6 ng/mL; normal range: 20.1~100 ng/mL). Subsequent cervical computed tomography (CT) scans displayed a well-defined enhancing mass measuring about 2.6cm in the retropharyngeal space at the upper level of the right thyroid cartilage. Based on clinical symptoms and radiological findings, additional parathyroid imaging was performed to precisely locate the suspected parathyroid adenoma. The parathyroid scan showed the presence of an upper right parathyroid adenoma in the retropharyngeal region, prompting the patient to seek surgical consultation (Figure 1). A transverse incision of approximately 5 cm was made slightly to the right of the midline of the neck, exposing the sternocleidomastoid (SCM) muscle. Following the lateral retraction of the SCM muscle, the right thyroid gland was discerned, and an incision was performed to reach the retropharyngeal space at the hyoid bone level. Enlarged lymph nodes were observed in the adjacent region, and by gently displacing the thyroid to the left, the parathyroid mass situated in the retropharyngeal space was revealed. Careful dissection was carried out to separate the retropharyngeal mass from the surrounding tissues. A portion of the excised mass was subjected to frozen section analysis, confirming its parathyroid origin. Intraoperative intact parathyroid hormone (iPTH) levels were measured and revealed a gradual reduction after the removal of the mass. The iPTH levels decreased to within the normal range as seen via serum testing 20 min after tumor removal. Lobectomy was performed through conventional techniques without injury to the recurrent laryngeal nerve and inferior parathyroid gland. No postoperative complications were observed, and the patient was discharged on the 5th day post-surgery. Subsequent pathology results of the thyroid nodule confirmed papillary thyroid cancer with no extrathyroidal extension or neural involvement and the retropharyngeal mass was diagnosed as a parathyroid adenoma (Figure 2). After the surgery, the patient’s blood tests showed that serum calcium (8.61 mg/dL), parathyroid hormone (32 pg/mL), and serum phosphate (4.57 mg/dL) levels had all returned to within the normal range. Ultimately, significant clinical improvement was achieved, and the patient is now under regular observation and follow-up (Figure 3).

pHPT is a common endocrine disorder characterized by the excessive secretion of parathyroid hormone, leading to hypercalcemia, which can manifest with symptoms such as osteoporosis, generalized weakness, and, in severe cases, altered consciousness. It occurs in approximately 1 in 500 women and 1 in 2000 men, and the primary treatment involves the complete removal of hyperfunctioning parathyroid tissue [1]. The majority of parathyroid tumors encountered in clinical practice are identified during the diagnostic workup for pHPT, which is characterized by hypercalcemia and elevated parathyroid hormone levels. Parathyroid adenomas exhibiting tissue invasion and lacking concurrent hyperparathyroidism are uncommon [2]. The correlation between end-stage renal disease and primary hyperparathyroidism is intricate, involving diverse underlying mechanisms. Continuous stimulation of both PTH synthesis and secretion occurs throughout the progression of chronic kidney disease, leading to the development of hyperparathyroidism. Downregulation of the vitamin D receptor and calcium-sensing receptor in parathyroid tissue further amplifies PTH overproduction. The persistent stimulation of parathyroid secretory function is marked not only by a gradual increase in serum PTH but also by the hyperplasia of parathyroid glands. This hyperplasia results from heightened parathyroid cell proliferation, insufficiently compensated by a concurrent increase in parathyroid cell apoptosis [3]. Additionally, primary hyperparathyroidism can be induced by causes such as parathyroid hyperplasia, multiple parathyroid adenomas, and parathyroid carcinoma [4]. Even in the absence of symptoms, the hyperfunction of the parathyroid glands should be treated with complete excision to prevent life-threatening arrhythmias and associated severe complications [5]. Therefore, precise localization of the parathyroid tumor along with accurate excision is crucial for the effective management of this condition [6].

The parathyroid glands exhibit considerable variation in their locations, and the superior parathyroid glands are commonly situated in proximity to the superior thyroid artery, constituting around 80% of cases. This description pertains to the localization of the superior parathyroid gland, and deviations in the position of the parathyroid glands may arise due to congenital displacement and other contributing factors. Variability also exists in the position and migration path between superior and inferior parathyroid glands, with superior parathyroid gland location changes being less frequent compared to the inferior parathyroid glands. These variations are attributed to embryological reasons. Surgery in the retropharyngeal space following the occurrence of parathyroid adenoma is challenging visually, as it is densely populated with the carotid artery and recurrent laryngeal nerve. It is crucial to exercise extreme caution during surgery to preserve the functionality of surrounding tissues.

Ectopic parathyroid glands are known to occur frequently in cases of recurrent hyperparathyroidism after initial parathyroidectomy [7]. However, in the present case, where there was no prior surgical history, the occurrence in the retropharyngeal space highlights the utility of technetium scanning for preoperative diagnosis. Localization of parathyroid tumors has employed various methods, including ultrasound, computed tomography, and nuclear imaging with technetium (Tc) 99m sestamibi. Neck ultrasound showed the least diagnostic accuracy at 84%, whereas computed tomography and Tc 99m scans presented higher accuracy rates of 92% and 90%, respectively. Isolated parathyroid adenomas have the potential to emerge at any anatomical site along their embryological descent pathway, with the mediastinum being the predominant location, constituting approximately 38%. In approximately 18% of instances, they may be localized within the thyroid gland itself. In uncommon cases, their origin may be traced to the parapharyngeal space [8].

The incidence of concurrent thyroid cancer and pHPT varies between 3.1% and 17%, and there is ongoing debate surrounding the underlying reasons for this correlation [9]. The coexistence of parathyroid adenoma and papillary thyroid carcinoma is an exceptionally rare phenomenon, with only a handful of cases reported in the existing literature. While certain authors attribute this simultaneous presence to chance, alternative researchers propose a connection to elevated endogenous calcium levels or potential involvement of growth factors like epithelial growth factors and insulin-like growth factors, suggesting their role as goitrogenic factors [10,11]. It is significant to highlight in this case that the incident occurred in the absence of any previous head and neck irradiation history. While the concurrent presence of thyroid cancer and parathyroid adenoma is infrequent, it is crucial to entertain the diagnosis of this co-occurrence in cases of pHPT to avert the necessity for reoperation, as exemplified in our patient’s case.

A comprehensive understanding of pHPT is imperative when considering surgery for parathyroid tumors, as is a thorough understanding of the anatomical location and variations in the parathyroid glands. Demonstrated in this case, the patient successfully achieved the clinical endpoint without complications through accurate assessment of the anatomical variation via appropriate diagnostic tests prior to surgical intervention. The emergence of a parathyroid adenoma in the retropharyngeal space emphasizes the need for thorough assessment to avoid misinterpretation as a distinct condition associated with thyroid cancer. Additionally, the assessment of anatomical variations through the application of appropriate diagnostic tools can prove beneficial in guiding decisions related to the clinical presentation, diagnostic approaches, surgical treatment, and surgical strategies for parathyroid tumors.

## Figures and Tables

**Figure 1 diagnostics-14-00110-f001:**
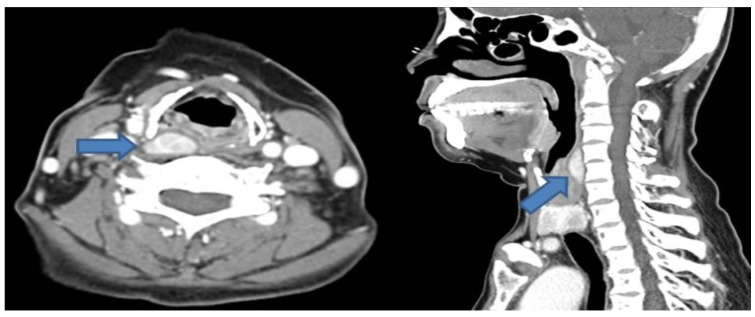
Preoperative enhanced neck CT shows an approximately 2.6 cm well-defined mass in the right retropharyngeal space (axial view, left blue arrow). The suspicious lesion in the retropharynx is located at the level of the upper thyroid cartilage (sagittal view, right blue arrow).

**Figure 2 diagnostics-14-00110-f002:**
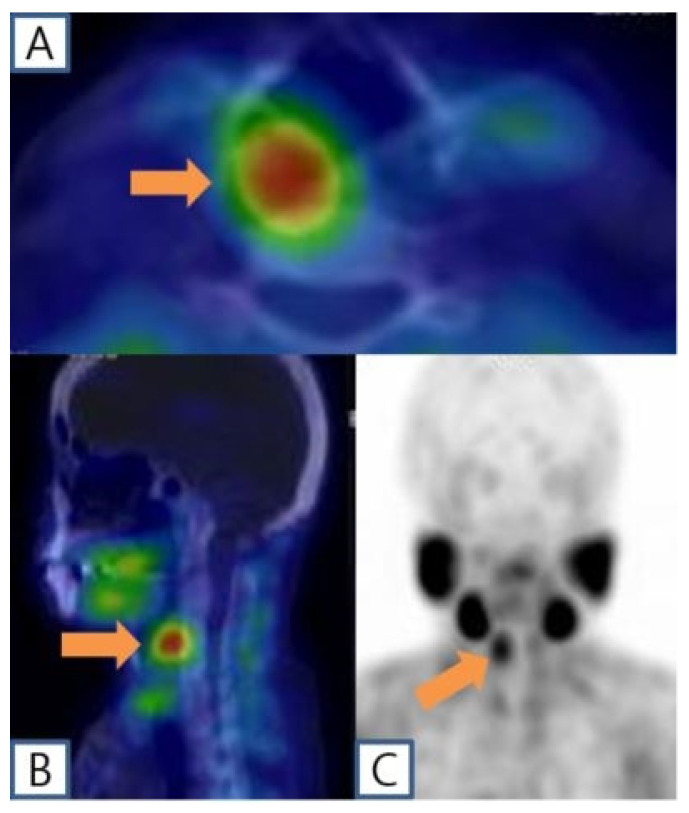
[99m Tc]Tc-MIBI single-photon emission computed tomography combining computed tomography (SPECT/CT) scintigraphy showing the right retropharyngeal lesion with sustained radiotracer uptake on the delayed images (**A**,**B**) (orange arrow). Maximum-intensity projection image (**C**) (orange arrow).

**Figure 3 diagnostics-14-00110-f003:**
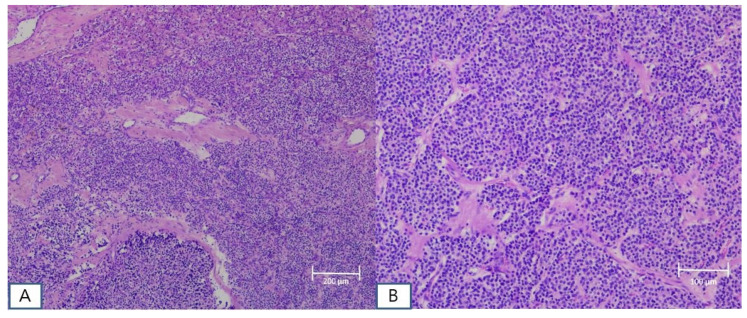
Histopathological examination confirmed the final diagnosis of parathyroid adenoma. Microscopic findings of the parathyroid gland show large cells with small irregular nuclei and dense acidophilic granules (H&E staining; original magnification, ×5) (**A**) and follicular structures predominantly composed of chief cells (H&E staining; original magnification, ×10) (**B**).

## Data Availability

The data presented in this study are available upon request from the corresponding author.

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
