# Peer review of "An Ectopic Parathyroid Adenoma of the Retropharynx in a Patient with Primary Hyperparathyroidism and Papillary Thyroid Cancer—A Rare Case"

_diagnostics, 2024, doi:10.3390/diagnostics14010110_

Round 1

Reviewer 1 Report

Comments and Suggestions for Authors

The article shares a rare case in terms of its content.

Some corrections are needed.

About the text in general:

*The summary can contain less information than the introduction and should include the patient's clinical outcomes.

*The article text is not divided from the figure description. In this state, its understandability has decreased. The main text should be separated by correcting the text's integrity and explaining the figure (e.g., figure 1).

*The two separate CT results are shown with two blue arrows in Figure 1, and the description should be written in more detail. The details shown with the arrows on the left and right should be clarified.

*A scale bar should be added to the H&E results in Figure 3.

*For the text to be reader-friendly, it should be divided into paragraphs.

*Sharing the patient's pre-op and post-op biochemical data is essential for the details to contribute to the literature. iPTH observed at normal levels during post-op is not a sufficient explanation; specific PTH, Ca, P, Vit D, etc., are not written in a simple Excel chart. Adding data will increase clarity.

Comments on the Quality of English Language

*I made a few English edits. I am sending it as an attached file. 

Reviewer 2 Report

Comments and Suggestions for Authors
